# A multiplex analysis of phonological and orthographic networks

**Pablo Lara-Martínez** [1], **Bibiana Obregón-Quintana**[1], **C. F. Reyes-Manzano**[2], **Irene López-Rodríguez**[3], **Lev Guzmán-Vargas** [3]*

**1** Facultad de Ciencias, Universidad Nacional Autónoma de México, Ciudad de México, México,
**2** Tecnológico Nacional de México, Tecnológico de Estudios Superiores de Ixtapaluca, Ixtapaluca, Estado de México, México, **3** Unidad Profesional Interdisciplinaria en Ingeniería y Tecnologías Avanzadas, Instituto Politécnico Nacional, Ciudad de México, México

* lguzmanv@ipn.mx

This is a Registered Report and may have an associated publication; please check the article page on the journal site for any related articles.

## Abstract

The study of natural language using a network approach has made it possible to characterize novel properties ranging from the level of individual words to phrases or sentences. A natural way to quantitatively evaluate similarities and differences between spoken and written language is by means of a multiplex network defined in terms of a similarity distance between words. Here, we use a multiplex representation of words based on orthographic or phonological similarity to evaluate their structure. We report that from the analysis of topological properties of networks, there are different levels of local and global similarity when comparing written vs. spoken structure across 12 natural languages from 4 language families. In particular, it is found that differences between the phonetic and written layers is markedly higher for French and English, while for the other languages analyzed, this separation is relatively smaller. We conclude that the multiplex approach allows us to explore additional properties of the interaction between spoken and written language.

## Introduction

The complexity of natural language has been studied from different perspectives of scientific research [1–5], among which characterizations based on phonological [6–8], morphological [9, 10], syntactic [11, 12], and semantic aspects [13, 14] stand out. Some of these approaches have shown that the complexity expressed in these aspects (phonetic, lexical, syntactic, semantic) are general properties, such as Zipf's law [15] and other linguistic laws, observed in all languages [1, 8, 16–18], while some particularities, such as the divergence between written and spoken language, may exhibit differences across languages. In some previous studies, the levels of complexity have been evaluated in terms of modeling based on complex single-layer networks or their extension to multilayer networks [19–21]. Of particular interest are findings about emergent organizational properties that encompass facets of language ranging from semantics to phonetics, including the written structure of language [22–25]. In many of these network-based approaches, it has been found that the behavior of the connectives -the number of neighbors of a given node- is often described by distributions that lie between power-law

**Data Availability Statement:** All corpora used in this study are available from the https://doi.org/10.6084/m9.figshare.14668593.

**Funding:** This work was partially supported by programs Estimulo al Desempeño de los Investigadores (EDI), Secretaría de Investigación y Posgrado (SIP-20221643) and Comisión de Fomento a las Actividades Académicas (COFAA) from Instituto Politécnico Nacional and Consejo Nacional de Ciencia y Tenología (CONACYT), México. We also acknowledge the partial financial support provided by DGAPA-UNAM, Mexico, through Grant No. IN111822. No additional external funding was received for this study. The funders had no role in study design, data collection and analysis, decision to publish, or preparation of the manuscript. The authors received no specific funding for this work.

**Competing interests:** The authors have declared that no competing interests exist.

and narrow exponential behavior, depending on the language and the association criterion. For instance, Arbesman et al. [6] reported that in the case of phonological networks, the degree distribution follows a truncated power law with different parameters when comparing different languages [6]. From the perspective of orthographic networks, it has been reported [22] that the distribution of connectives for the mental lexicon of elementary-level learners is well described by a power law with small-world properties. Although different natural language properties based on transformations to complex networks have been analyzed, few of them have focused on incorporating multilayer aspects of the language [20, 26]. In this study, we address orthographic and phonetic features of language using a multiplex approach. In particular, by estimating the similarity between word pairs, a two-layer network is constructed in which the nodes are the words, and a link exists if a threshold value of the similarity is satisfied. For the purpose of our study, this distance similarity between two words, A and B, can be defined as the minimum number of edit operations needed to transform A into B, which is the well-known Damerau-Levenshtein (DL) [27–29] (see Methods). Among the widely recognized characteristics of many natural languages is the non-existence of a biunivocal correspondence between the writing of a word and its corresponding pronunciation. Thus, the correspondence between graphemes and phonemes is not biunivocal, giving rise to situations such as homography (when one letter corresponds to two phonemes), digraphy (two letters correspond to one phoneme or vice versa), heterography (one phoneme corresponds to two or more letters), etc. [30–35]. In fact, at the word level, the appearance of phenomena such as homophony and transparency in natural languages has been the subject of extensive study from the linguistic perspective [36, 37]. On the other hand, the use of complex networks has been incorporated into systems analysis as the language where multiplex modeling is most appropriate. In these cases, the nodes are placed in layers with connections between them and the nodes are common to all layered networks. Several real and simulated multilayer networks have been studied in contexts such as finance and economics [38–40], social systems [41, 42], synchronization [43] and linguistics [21]. A direct comparison between orthographical and phonological networks would be important to quantify the local and global connectivity patterns and their changes across different languages. Related to the latter, and in the context of psycholinguistic studies, the identification of these differences and similarities potentially contribute to the understanding of the mechanisms that act on cognitive processes, such as word recognition and retrieval, and whose manifestations are particularly different when looking at orthographic or phonetic organization. In a more potentially applicable context, the relationship between orthographic and phonological networks could be of great interest for the robustness of automatic speech recognition systems, as they are often prone to failures in transcription to written text. And it could also impact issues such as cross-language transfer learning, where a neural network that can recognize one language might perform well in another language depending on the similarity of the multiplexed network. The contributions of this study focus on answering the questions that were posed in the registered protocol and are summarized as follows. (i) Unlike previous studies based on single-layer networks, multiplex orthographic-phonetic networks were constructed for 12 natural languages based on similarities between $5\times10^4$ words. The observed properties reveal that it is possible to differentiate levels of organization between orthographic and phonetic structure in natural language. (ii) Our results indicate that while certain languages exhibit a high correlation, for node-based measures, between phonetic and orthographic similarity, for others this correlation is rather low, reinforcing the identification of differences at the local level. (iii) Our approach based on a multiplex analysis presents an alternative view for understanding the organization of language by combining the written and spoken form.

## Orthographic and phonological network

In this study, the multiplex language network consists of an orthographic network and phonological network (see Fig 1 for a schematic representation). For the orthographic network, we generate a similarity network at word-level $G^{[O]} = (V^{[O]}, E^{[O]})$, where nodes are words and a link between two nodes is defined if the DL distance is smaller or equal than a threshold value $\ell$. In a similar way, the phonological network $G^{[P]} = (V^{[P]}, E^{[P]})$ is constructed in terms of nodes which represent words translated to the international phonological alphabet (IPA), and links are defined if the DL, is smaller or equal than a given threshold $\ell$. Next, the orthographic and phonological networks are combined to generate a two-layer network, denoted by $G_L^{[\alpha]} = (V^{[\alpha]}, E^{[\alpha]})$, with $\alpha = O, P$. Here, the adjacency matrix for the multiplex network is given $a_{ij}^{[\alpha]}$, where $a_{ij}^{[\alpha]} = 1$ indicates that there is a link between node (word) $i$ and node (word) $j$ at layer $\alpha$. More formally, the adjacency matrix associated with each layer is defined as: $a_{ij}^{[\alpha]} = [\Theta(\ell - d_{ij}^{[\alpha]}) - \delta_{ij}]I_{max}^{[\alpha]}(i,j)$, where $\Theta(-)$ represents the Heaviside function, $\delta_{ij}$ is the Kronecker delta, $d_{ij}^{[\alpha]}$ the DL distance between word $i$ and word $j$ at layer $\alpha$. Here, the factor $I_{max}^{[\alpha]}(i,j) = [1 - \Theta(d_{ij}^{[\alpha]} - \max(\lambda_i^{[\alpha]}, \lambda_j^{[\alpha]})]$ is considered to exclude the cases for which the link does not reflect similarity, and $\lambda_i^{[\alpha]}$ and $\lambda_j^{[\alpha]}$ are the lengths (in characters) of words $i$ and $j$, respectively.

## Databases

Our study focuses on analyzing orthographic-phonological networks of 12 natural languages belonging to four language families: Germanic (English, German, Dutch and Swedish), Romance (French, Spanish, Portuguese and Italian), Slavic (Russian, Ukrainian and Polish) and Uralic (Hungarian). A corpus of words for each language was constructed using a set of books available from the Gutenberg project www.gutenberg.org. The written texts were pre-processed to remove function words, stop words and any mark symbol. The titles of the written texts and the resulting corpus are described in https://doi.org/10.6084/m9.figshare.14668593 [44]. The final corpora contains $50 \times 10^3$ words with their corresponding translation

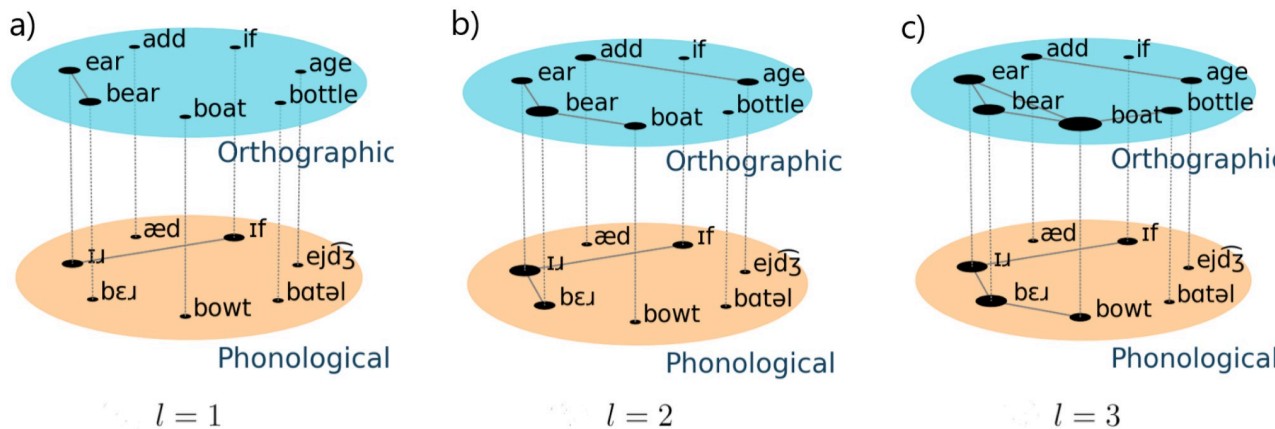

**Fig 1. Construction of the multiplex language network.** Representative multiplex network for English language and several distance thresholds. Each layer represents the orthographic (top) and phonological (bottom) networks. Here, nodes are words and there is a link if the Damerau-Levenshtein distance is smaller than a given threshold (a) $\ell = 1$, (b) $\ell = 2$ and (c) $\ell = 3$. Notice that words in the phonological layer were translated into the International Phonetic Alphabet and then the DL was calculated. The figures where generated by using the free python library *pymnet* [45, 46].

to the international phonetic alphabet for each language (transliterated by the epitran library of Python version 3.6.8).

## Results

First, prior to the description of the multiplex properties of the OP network, we present updated results for the calculations of the basic structural properties of the orthographic and phonological networks (see Methods for details). These results correspond to topological features calculated for corpus with $50 \times 10^3$ words for each of the 12 languages in our study. Table 1 shows the results for 4 representative languages of each linguistic family (see supporting information online at FigShare [44] for results from all the languages in our study). For a comparison between the interlayer values of these network metrics, the phonological/orthographic ratios for the different languages are shown in Fig 2.

**Table 1. Results for the basic topological network quantities obtained from the ortographic ($G^O$) and phonological ($G^P$) networks.**

| Language | Metric / Network | $G^O$ | | | $G^P$ | | |
|---|---|---|---|---|---|---|---|
| | Threshold | $\ell = 1$ | $\ell = 2$ | $\ell = 3$ | $\ell = 1$ | $\ell = 2$ | $\ell = 3$ |
| English | Density | $1.38(10^{-4})$ | $10.32(10^{-4})$ | $74.58(10^{-4})$ | $2.62(10^{-4})$ | $18.05(10^{-4})$ | $97.81(10^{-4})$ |
| | Average degree $\bar{k}$ | 4.76 | 47.57 | 366.41 | 8.79 | 81.63 | 476.86 |
| | Nearest neighbor $\bar{k}_{nn}$ | 5.90 | 66.94 | 552.60 | 10.92 | 113.03 | 689.39 |
| | Clustering $\bar{c}$ | 0.20 | 0.28 | 0.32 | 0.21 | 0.31 | 0.35 |
| | Average component size | 3.63 | 18.58 | 143.69 | 4.49 | 19.29 | 125.42 |
| | Maximum modularity $Q^*$ | 0.80 | 0.55 | 0.39 | 0.80 | 0.55 | 0.39 |
| Spanish | Density | $0.82(10^{-4})$ | $4.66(10^{-4})$ | $33.46(10^{-4})$ | $1.02(10^{-4})$ | $6.69(10^{-4})$ | $47.95(10^{-4})$ |
| | Average degree $\bar{k}$ | 2.79 | 21.19 | 163.83 | 3.67 | 31.17 | 235.96 |
| | Nearest neighbor $\bar{k}_{nn}$ | 3.43 | 29.65 | 249.16 | 4.58 | 43.77 | 354.44 |
| | Clustering $\bar{c}$ | 0.14 | 0.31 | 0.31 | 0.14 | 0.30 | 0.31 |
| | Average component size | 2.75 | 16.00 | 118.23 | 3.36 | 23.26 | 161.59 |
| | Maximum modularity $Q^*$ | 0.85 | 0.54 | 0.38 | 0.85 | 0.54 | 0.38 |
| Russian | Density | $0.82(10^{-4})$ | $2.90(10^{-4})$ | $19.96(10^{-4})$ | $0.86(10^{-4})$ | $2.57(10^{-4})$ | $14.48(10^{-4})$ |
| | Average degree $\bar{k}$ | 2.21 | 12.61 | 95.69 | 2.16 | 10.77 | 68.16 |
| | Nearest neighbor $\bar{k}_{nn}$ | 2.66 | 17.26 | 145.03 | 2.60 | 14.60 | 101.85 |
| | Clustering $\bar{c}$ | 0.22 | 0.34 | 0.32 | 0.21 | 0.35 | 0.34 |
| | Average component size | 2.28 | 9.25 | 49.25 | 2.19 | 7.14 | 28.47 |
| | Maximum modularity $Q^*$ | 0.95 | 0.71 | 0.49 | 0.95 | 0.71 | 0.49 |
| Hungarian | Density | $1.08(10^{-4})$ | $3.26(10^{-4})$ | $18.19(10^{-4})$ | $1.13(10^{-4})$ | $3.28(10^{-4})$ | $16.40(10^{-4})$ |
| | Average degree $\bar{k}$ | 2.36 | 12.77 | 83.07 | 2.36 | 12.39 | 73.43 |
| | Nearest neighbor $\bar{k}_{nn}$ | 2.92 | 17.74 | 129.16 | 2.90 | 16.81 | 109.21 |
| | Clustering $\bar{c}$ | 0.17 | 0.31 | 0.35 | 0.18 | 0.30 | 0.33 |
| | Average component size | 2.31 | 9.18 | 34.88 | 2.30 | 8.18 | 30.56 |
| | Maximum modularity $Q^*$ | 0.94 | 0.72 | 0.54 | 0.94 | 0.72 | 0.54 |

Table notes. Topological metrics of the orthographic network and the phonological network. These results were obtained from networks with $50 \times 10^3$ words at each layer. The average values of the degree ($\bar{k}$), clustering ($\bar{c}$) and nearest neighbor ($\bar{k}_{nn}$) are presented. We observe that the density, $\bar{k}$, $\bar{c}$ and $\bar{k}_{nn}$ exhibit an increasing behavior for the four languages and the two layers, with some similarities such as it occurs for $\bar{c}$ in both layers and distances $\ell = 2, 3$. For the modularity and the average cluster size, we observe they exhibit opposite trends, while the modularity decreases as $l$ increases, the average cluster size increases because a larger number of nodes tends to be connected to a giant component. See extended data online at https://doi.org/10.6084/m9.figshare.14668593 FigShare [44].

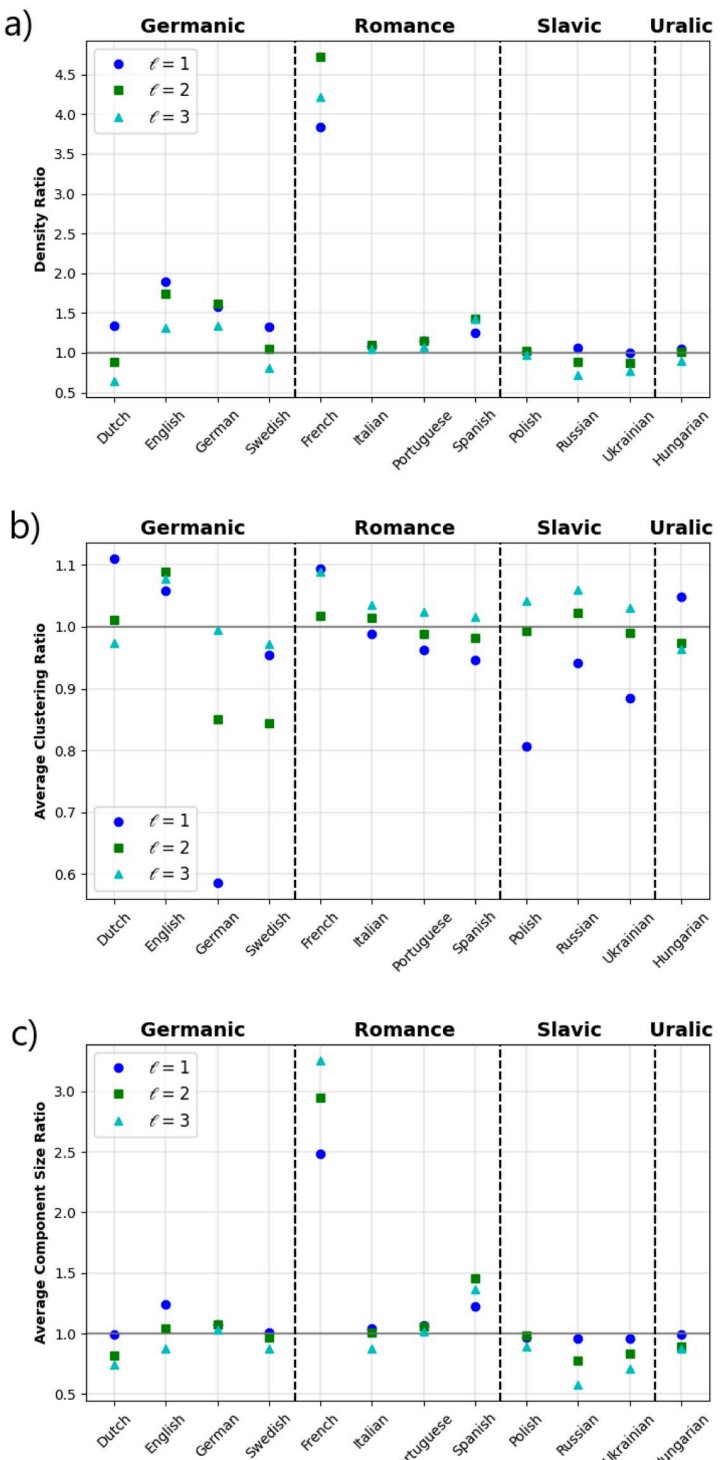

**Fig 2. Ratio (phonological/orthographic) values for some metrics shown in Table 1.** The cases of (a) density, (b) average clustering, and (c) average component size are depicted. Here a value close to 1 indicates that similar metric-values are obtained either for the ortographic or phonological layer, while a value greater (smaller) than 1 is obtained when the phonological (orthographic) exceeds the opposite layer. It is observed that French exhibits the highest asymmetry for density and component size, while for clustering, most languages display values close to 1, except German with higher values in the orthographic layer.

The density ratio (Fig 2a) indicates that the phonological network has more connections than the orthographic network, confirming that the sound affinity between words is greatest in languages such as French, and to a lesser extent for English and German. This result also aligns with previous findings about properties like homophony (when two or more words sound the same, but carry distinct meanings) in several human languages [33], which favors the increase of the degrees in the phonetic layer, while the word spelling is circumscribed by the repetition of the characters. The average clustering coefficient exhibits relatively similar ratio values for almost all languages (Fig 2b), indicating that the local structure (presence of triangles) is similar whether looking at orthographic or phonological properties. Moreover, as shown in Fig 2, the average cluster size obtains larger values in the phonological vs. orthographic layer for French and to a lesser extent for Spanish, while for the rest of the languages the difference is smaller and even reverses this behavior for the Slavic family and Hungarian. This reveals that the fragmentation of the layers occurs differently when comparing the languages, with the phonological layer showing the greatest cohesion in the languages noted above.

In addition, the determination of the functional form of the degree distribution of nodes has gained notoriety for establishing the behavior of connectivities and structural analysis of networks [47]. In the case of the distributions corresponding to the orthographic and phonological layer, it is observed that they correspond to distributions with a broad degree distribution, also known as fat-tailed distribution [48] (See Fig 3). For each degree distribution in our study, we performed fits to the data by considering the following distributions: Gumbel, Exponential, Loglogistic, Lognormal, Weibull and Power-law (see Supplementary Material [44]). To establish the best distribution that fits the data of the phonological and orthographic networks, we used the Akaike and Bayesian information criteria [49, 50] (see Methods and Supplementary Material [44] for details). According to statistical tests, most of the degree distribution networks of each layer (either orthographic or phonological) can be well described by the Weibull distribution, while for the remaining distributions the best fits correspond to

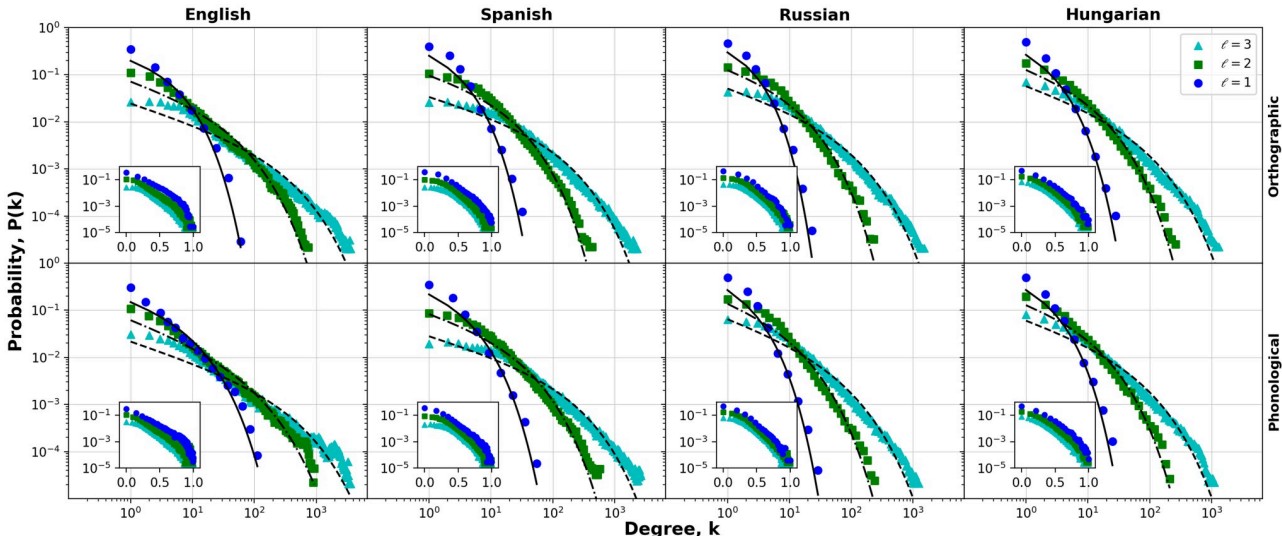

**Fig 3. Degree distributions for phonological and orthographic networks.** The cases of English, Spanish, Russian and Hungarian are depicted for three distance thresholds. The top row shows the distributions of the orthographic layer, while the bottom row shows the phonological layer. Dashed lines represent a Weibull-type function fit (see Table 4 in Supplementary Material at FigShare [44]). We find that for the majority of languages and both layers, the distributions display a heavy-tailed behavior. For a better comparison of the data, the insets of each plot show the corresponding degree distribution for normalized degrees $k/k^*$ (horizontal axis), where $k^* = \max(\log(k))$.

the Loglogistic and Lognormal, although the Weibull remains the second best fit in most of these cases (see Supplementary Material [44]). The corresponding survival cumulative distribution of the Weibull is represented by a stretched exponential function which have been used to describe a variety of phenomena [51–56]. We note that this distribution is more skewed than a single exponential distribution but less skewed than a power law distribution. As shown in Table 4 of the Supplementary Material [44], the Weibull fitting exponents ($\hat{\alpha}$ and $\hat{\lambda}$, for the orthographic are slightly greater than the corresponding phonological exponent, except for Spanish and French, indicating that larger connectivities are present in phonological networks, i. e., a relatively small number of words concentrate similarity with many other words in terms of phonetic structure. These new values for the scaling exponents improve the description of the connectivities previously reported in the preliminary analysis [57], and are consistent with the fact that the distributions are of the heavy-tailed type. To reinforce the choice of the fit to a Weibull function in most cases, we have performed the likelihood ratio test as described by Clauset et al. [58] (see Methods for details). The results show that almost all the fits lead to $p$-values lower than $10^{-10}$, in terms of the probability that they fit better to a Weibull-type function than to any other of the four distributions considered in our analysis. Similar conditions were found when considering cases where the selection corresponds to Loglogistic or Lognormal. Our findings of the behavior of degree distributions are also in general agreement with previous estimates made for some languages and based on phonological and written similarities [6].

In order to assess the similarities between distributions from different languages we resort to a robust measure to estimate the distance between them: the Jensen-Shannon distance (JSD), (see Methods). Fig 4 shows the matrix of JSD values between all pairs of orthographic vs phonological degree distributions for $\ell = 1, 2, 3$. The order of columns and rows has been determined by their similarities by using an agglomerative hierarchical clustering methodology (see Methods for details). The resulting dendograms are shown in top and left sides of the distance matrices. From the orthographic perspective, it is observed that English is the most divergent from the rest, while Russian, Hungarian and Ukranian are the least distant to one

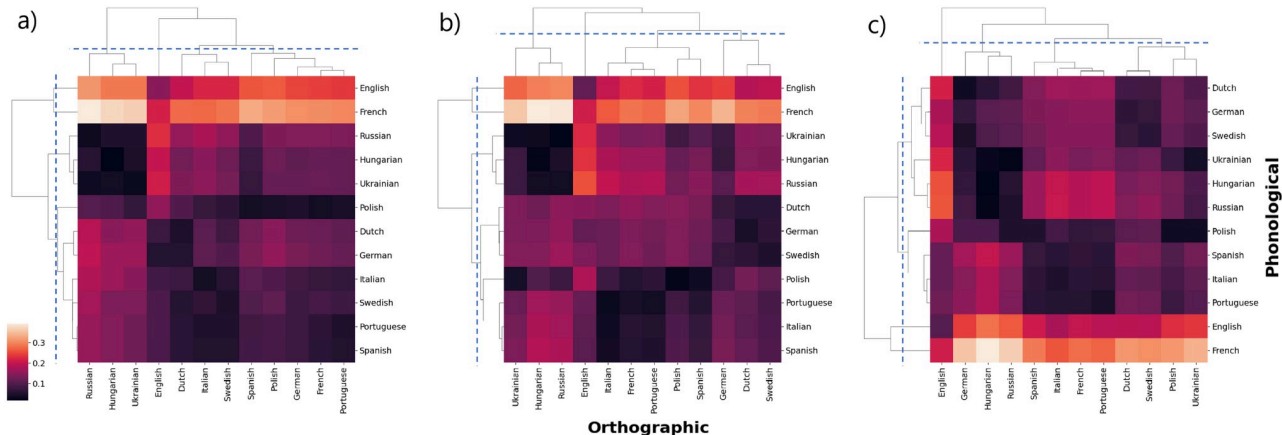

**Fig 4. Language similarity evaluated by the Jensen-Shannon distance between layers orthographic (horizontal) and phonological (vertical).** The cases of (a) $\ell = 1$, (b) $\ell = 2$ and (c) $\ell = 3$ are depicted. The dendrograms have been determined in terms of the similarities between languages by using the agglomerative hierarquical clustering method. We observe that for $\ell = 1$ and $\ell = 2$, the dendogram for the orthographic dimension at intermediate height (dashed line), four groups are identified, G1 (Russian, Hungarian and Ukranian) is the one which exhibits the highest internal similarity (low JSD); the other three groups correspond to G2 (English), G3 (Dutch, Italian and Swedish), G4 (Spanish, Polish, German and Portuguese). It is important to notice that in groups G3 and G4, Romance, Germanic and Slavic families are mixed and English is an isolated language. In contrast, for the phonological dimension, the JSD values at an intermediate cut-off (dashed line), also four groups are again observed, being the English and French the ones that stands out for a large distance with any other language, while Ukranian, Russian and Hungarian are described by relatively low distances. For $\ell = 3$, we observe that English is the most divergent from the rest in terms of writing, while English and French are the most divergent in terms of phonological structure.

another, specially for $\ell = 1$ and $\ell = 2$. With respect to the phonetic component, clearly English and French are the languages with the greatest separation from the rest, while Russian, Hungarian and Ukrainian are also the closest languages.

Next we further explore the relationship between some topological features of the orthographic and phonetic networks. First, the Spearman-rank coefficient is calculated to evaluate the presence of correlations. Fig 5 shows the results of the calculations of the correlations for degree, clustering and average nearest neighbor. Positive correlations are observed in all properties, but some differences are remarkable when comparing individual languages and linguistic families. For degrees (Fig 5a), we observe that English, German and French exhibit a relatively low correlation values for the threshold value $\ell = 3$, while for the rest of languages and the three threshold values, a higher correlation is present ($\geq 0.7$). These results indicate that, for most of the languages in our study, words with high similarity in their orthographic structure tend to have also more phonological similarity and viceversa, except for the three languages listed above. For $k_{nn}$ (Fig 5b), higher and similar coefficient values for the correlation are observed for all languages and threshold values $\ell = 1$ and $\ell = 2$, confirming that by increasing the mean spelling similarity (with other words) of the neighbors of a given word, the phonetic similarity of the neighbors also increases. This fact is particularly remarkable for Romance languages, except French. For clustering (Fig 5c), languages which belong to the Germanic family and French have lower correlations coefficients, revealing that words with a high fraction of connected (with similar orthographic structure) neighbors tend to have rather a smaller fraction of connected neighbors in terms of phonological similarity and viceversa.

In other to evaluate the link overlap across orthographic and phonological layers, the normalized local Jaccard's index was calculated (see Methods). Here, a value close to one would indicate that words tend to have the same neighbors, either in the orthographic or phonological layer, while when it is close to zero, words do not necessarily share the same neighbors. Fig 6a shows the results of the calculations of the Jaccard's index. The Germanic family (Dutch, English, German and Swedish) together with French display a relatively low index value, indicating that a low overlapping is present across both layers, while for the Romance, Slavic and Uralic families, similar values are observed which represent the fact that words tend to have the same neighbors across layers.

Moreover, we also explored the similarities between both layers from the perspective of modularity, which measures the property of a given network to be divided into groups [59]. First, we evaluated the ratio between single-layer modularity. Fig 6b show the ratio phonological/orthographic for all languages. It is observed that for Dutch and Swedish the dissociation between modules tends to be markedly high in the phonetic layer compared to the orthographic one, while the opposite is observed for French, i.e., for French the modules tend to be not very well defined due to a dense connectivity with different phonetically similar groups. Next, to get further insight in the identification of differences and common properties between layers, we explore the similarity between the communities associated to a given modularity $Q$, i.e. to which extent words which belong to a certain community in the orthographic layer are also contained in the corresponding group in the phonological network. We resort to the index $F1^{*}$-score (see Methods) to estimate the similarity (in terms of overlapping of communities) between the two layers. Fig 6c shows that for English and French $F1^{*}$ exhibits lower values, specially for $\ell = 3$, indicating that for these languages words tend to fall into different communities across layers. In contrast, higher values for $F1^{*}$ are observed for the rest of the languages, which are consistent with the fact that most words are identified with the same group regardless of whether they are written or spoken.

Finally, we performed multiple robustness tests to explore the similarities and differences between layers in terms of two general strategies: directed attacks and random failures [60]. To

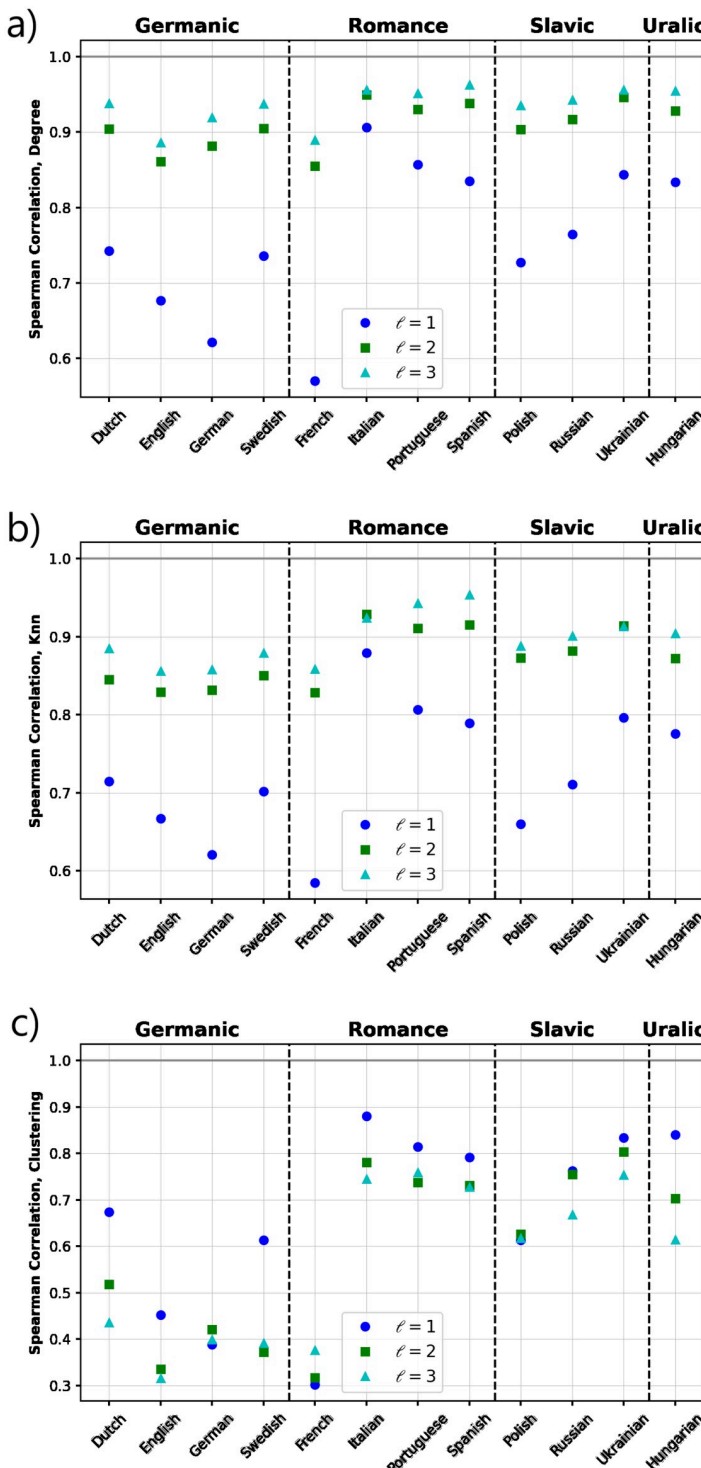

**Fig 5. Correlations between some structural properties of the orthographic and phonological networks.** We show the Spearman-rank correlation coefficient for (a) degree, (b) average nearest neighbor degree and (c) average clustering. For most of the languages, similar levels of positive correlations are observed for the three properties, except the cases of the Germanic family and French for which the clustering is noticeable lower compared to the rest of languages.

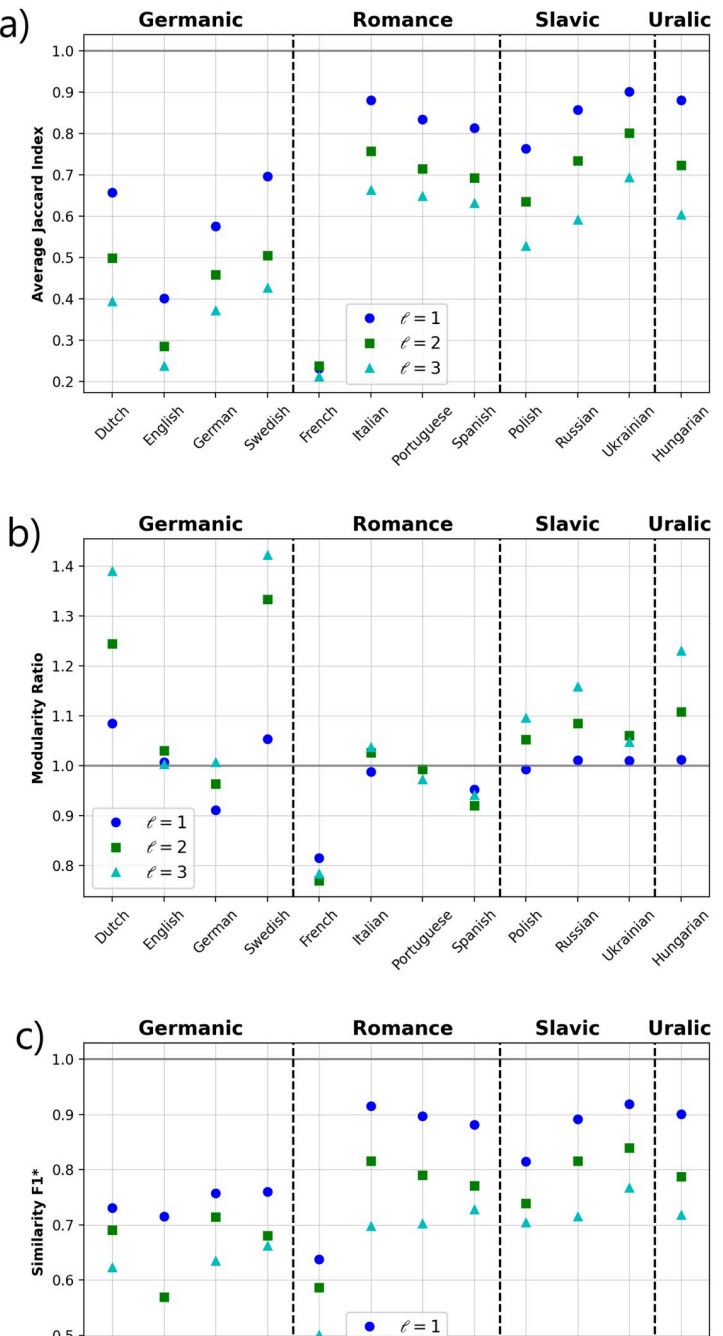

**Fig 6. Average Jaccard index ($J$), modularity ratio and similarity $F1^*$.** (a) Jacccad's index which indicate the extent of link overlap across layers. (b) Modularity ratio between layers. The value of the phonological layer divided by the value of the orthographic layer is shown. (c) Score $F1^*$ to evaluate the similarity between communities associated to a given modularity.

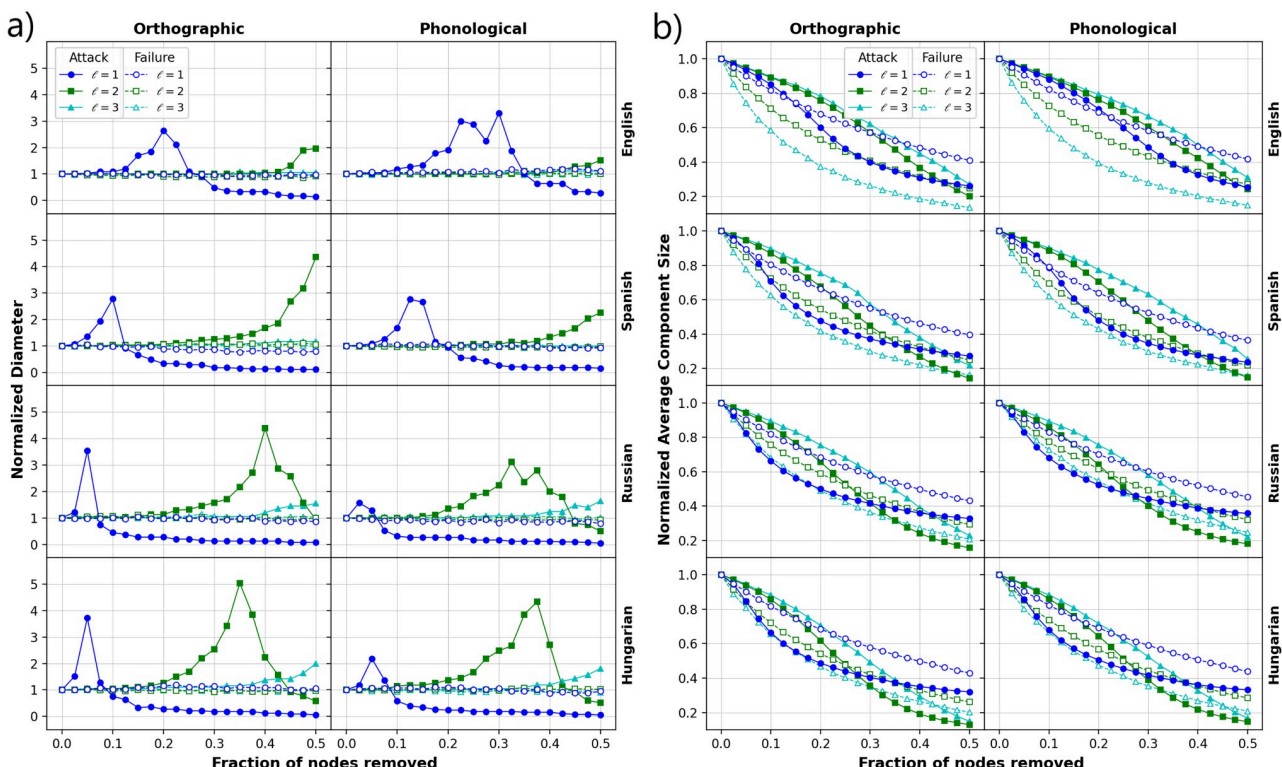

**Fig 7. Robustness of the networks.** Directed attacks consist in removing the most connected nodes and for the failures nodes are removed randomly. a) Behavior of the normalized diameter in terms of the fraction of removed nodes. b) As above but for normalized component size. Different profiles for the decay are observed when comparing orthographic and phonological networks from the four languages. The results for failures correspond to the average from 10 independent realizations.

test whether both layers are affected by the removal of a fraction $f$ of most connected nodes or selected at random, the mean size of the components and the diameter of the networks were monitored. Fig 7 presents the results of the calculations for four representative languages (see extended data online at Figshare [44]). For attacks, we find that for $\ell = 1$ and for both layers (Fig 7a), as we increase the fraction of removed nodes, the normalized diameter tends to increase until a maximum value and then it decreases. The threshold value ($f^*$) for which the diameter exhibits a peak changes for each language, while for Slavic languages and Hungarian is located below 0.1, for Germanic and Romance languages it is located slightly above 0.1 (see accompanying information online at FigShare [44]). Interestingly, this transition occurs in the phonological layer systematically for slightly larger values of removed node fractions, except for the Slavic family and Hungarian, which exhibit transitions at similar values of $f$. For larger values of the DL distance, the transition point seems to be located to the right, i.e., a larger value of $f$ is needed to detect the transition (Fig 7a). For the case of random failures, increasing fractions of removed nodes reveals a limited effect on the normalized diameter in all languages. In contrast, when the average component size is monitored in terms of $f$, important differences emerge between the layers and languages (Fig 7b). We observe that both layers exhibit a decaying behavior with different rates for attack compared with random failures. For a more direct comparison between both layers, we computed the average differences between the values of the normalized component size of the phonetic and the corresponding values of the orthographic layer for either attack or random failures (Fig 8). French is the language with the largest deviations either positive or negative, i. e., the orthographic layer is more robust under attacks while for random failures, the phonological layer is more robust. A similar behavior,

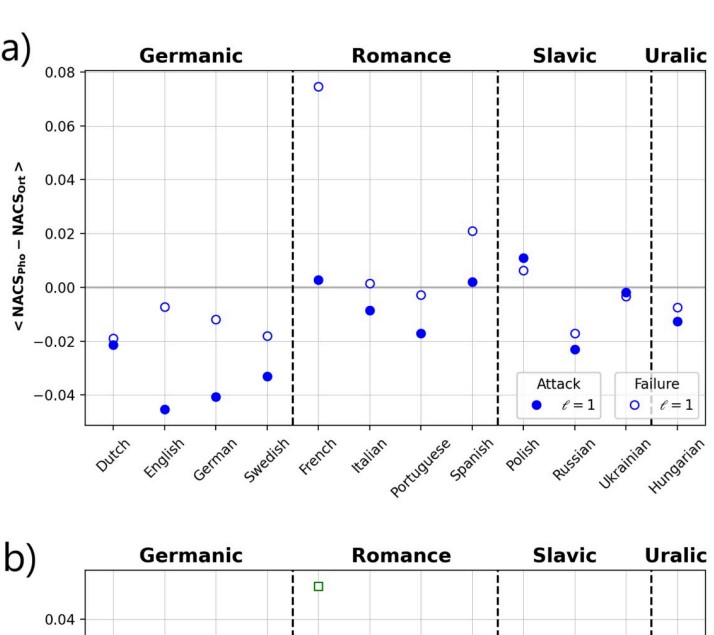

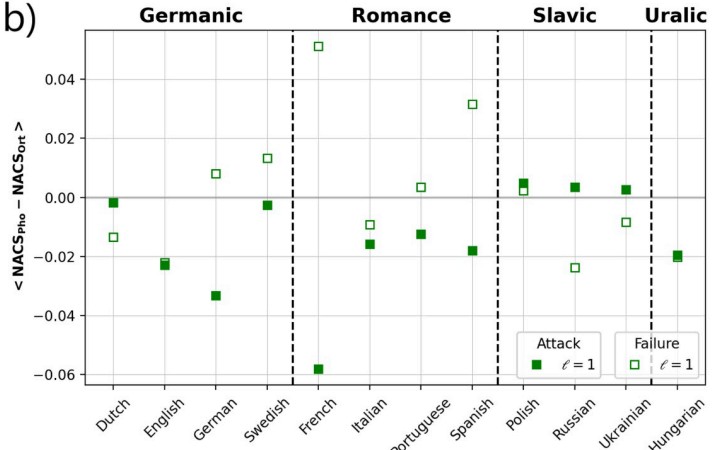

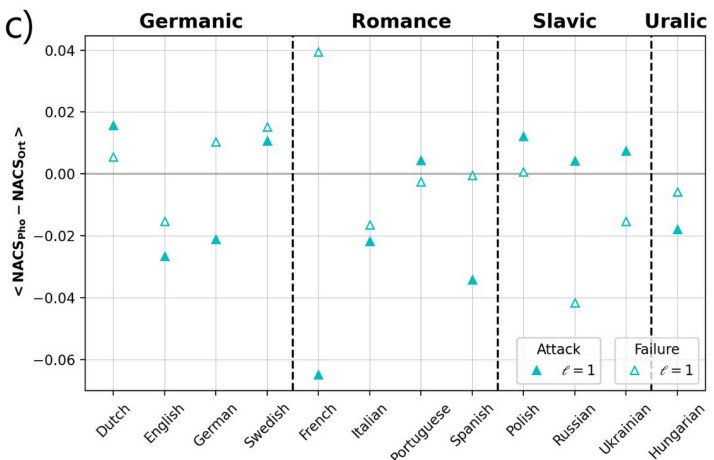

**Fig 8. Average component size differences between phonological and orthographic networks.** We show the cases of attacks (filled circles) and failures (open circles) for several thresholds $\ell$ of the DL distances, a) $\ell = 1$, b) $\ell = 2$, and c) $\ell = 3$. The results for failures correspond to the average from 10 independent realizations.

but to a lesser extent, is present in Spanish, while for Germanic languages a greater robustness is observed from the orthographic perspective. The rest of the languages exhibit small deviations with a slightly higher orthographic robustness either for failures or attacks.

## Discussion and conclusion

We have shown that orthographic and phonological layers exhibit similarities and differences across several languages from four linguistic families. Interestingly, our network analysis based on a wide variety of measures showed that natural languages reveal different levels of proximity when viewed from the written or spoken perspective. Our findings significantly extend previous studies based on small word corpora and limited to a few languages [32]. In previous works, models of inter-word similarity networks have been approached based only on purely orthographic or phonological properties, showing that there are changes in network characteristics when different languages are compared. In our approach, the different metrics are compared in parallel and the differences or asymmetries are highlighted across languages. More importantly, the results about a higher density in the phonological layer for languages like French and English are consistent with linguistic reports which point out the presence of homophony and more opacity in these languages [31, 33, 36, 37].

A remarkable fact, in the context of our study, is that when languages were grouped based on the distance between connectivity distributions, the categorizations did not necessarily correspond to the classification by language family, with cases such as English and French having a greater divergence (specially phonetic) with respect to the rest of the languages. Although our approach is based on a simple string-distance metric without incorporating other elements such as syllables, morphemes, etc., the similarities and differences suggest that additional quantitative evaluations, which can incorporate these additional information, can be performed across several natural languages. The results we report here are in general agreement with studies focused either on phonological or orthographic networks, and reinforce the idea of common general organization in natural languages. In addition, the accuracy of the network metrics and their changes across layers to determine similarities and differences make it appropriate to benchmark with other languages, and eventually apply this approach on a scale beyond the word level. The present study can be naturally extended with the incorporation of additional layers containing semantic information, polarity information, etc., to explore additional properties with potential use in contexts of text classification, automatic speech recognition systems and pattern identification in natural languages.

Our study has some limitations, the most notorious of which is that the similarity of phonetic structure based on the Damerau- Levenhstein distance tends to be overestimated because the discretization of sounds leads to a loss of structure [8]. Also, the sample size may impact the estimation of some parameters of the multiplex network.

We conclude that the multiplex analysis reveals additional features, which have not been evaluated by other methods, and provides a way to obtain important information about the interaction between spoken and written language. In addition, this study offers an alternative multiplex network-based methodology for language analysis and can be easily extended to other languages to contribute to the understanding of language complexity.

## Methods

### Damerau-Levenshtein distance

The distance similarity between two strings A and B can be defined as the minimum number of edit operations needed to transform A into B. These operations are: (1) substitute a character in A to a different character, (2) insert a character into A, (3) delete a character of A, and

(4) transpose two adjacent characters of A. The Damerau-Levenshtein (DL) distance is then defined as the length of the optimal edit sequence [27–29]. In our analysis, we adopt the DL distance $\ell$ as a threshold value to define a link between two words.

## Network metrics

Our analysis is focused on the basic topological characteristics of individual networks, and then to proceed to investigate similarities and differences of the two layers. We listed the single-layer-network measures (of a network with $N$ nodes) in a multiplex network that are evaluated [61, 62]:

- Density. The density of a layer $\alpha$, $\rho^{[\alpha]}$, is given as:

$$\rho^{[\alpha]} = \frac{2m^{[\alpha]}}{N(N-1)} \tag{1}$$

where $m^{[\alpha]}$ is the number of actual connections within the layer $\alpha$.

- Degree distribution. The degree $k_i^{[\alpha]}$ of a node $i$ is the number of links outgoing (or incoming) to that node,

$$k_i^{[\alpha]} = \sum_{j=1}^{N} a_{ij}^{[\alpha]}. \tag{2}$$

The degree distribution for layer $\alpha$ is then defined as the fraction of nodes in the network with degree $k$,

$$P^{[\alpha]}(k) = \frac{n_k^{[\alpha]}}{N}, \tag{3}$$

where $n_k^{[\alpha]}$ is the number of nodes with degree $k$.

- Clustering Coefficient. Measures the degree of transitivity in connectivity among the nearest neighbors of a node $i$ within the layer $\alpha$. $C_i^{[\alpha]}$ is calculated as [61],

$$C_i^{[\alpha]} = \frac{2E_i^{[\alpha]}}{k_i^{[\alpha]}(k_i^{[\alpha]} - 1)}, \tag{4}$$

where $E_i^{[\alpha]}$ is the number of links between the $k_i^{[\alpha]}$ neighbors of the node $i$ within the layer $\alpha$.

- Average Nearest-Neighbor Degree. Measures the average of the neighbors of a node [61]. The $\bar{k}_{nn,i}^{[\alpha]}$ is calculated as:

$$\bar{k}_{nn,i}^{[\alpha]} = \frac{1}{k_i^{[\alpha]}} \sum_{j=1}^{N} a_{ij}^{[\alpha]} k_j^{[\alpha]}. \tag{5}$$

- Modularity. Given $c_i^{[\alpha]}$ the community associated to the node $i$ within the layer $\alpha$, where $c_i^{[\alpha]} \in \{1, 2, \ldots, P\}$, with $P$ a natural number. The modularity, $Q^{[\alpha]}$ of a given layer $\alpha$ is given by [62]:

$$\mathcal{Q}^{[\alpha]} = \frac{1}{2m^{[\alpha]}} \sum_{ij} \left( a_{ij}^{[\alpha]} - \frac{k_i^{[\alpha]} k_j^{[\alpha]}}{2m^{[\alpha]}} \right) \delta(c_i^{[\alpha]}, c_j^{[\alpha]}), \tag{6}$$

where $\delta$ is the Kronecker delta. We use the Louvain algorithm [63] to perform a greedy optimization of the modularity.

## Fitting of degree distributions

To determine functional forms of the degree distributions, we have resorted to procedures based on two indicators: the Akaike information criterion (AIC) [50, 64] and the Bayesian information criterion (BIC) [49, 50]. These criteria represent two of the most widely used families of model selection indicators for identifying the "best model". Details of our procedure for discriminating the significance of adjustments can be found in the Supplementary Material [44]. Besides, we evaluated the goodness-of-fit by calculating the $p$-values of the likelihood ratio test introduced by Clauset et al. [58, 65] to compare the fits, thereby confirming that in most the cases, the observed data fit better to a Weibull distribution than to any of the other four distributions considered in our analysis. Similarly, when the best fit corresponded to Loglogistic or Lognormal, the $p$-values were very small ($p < 10^{-10}$)

## Jensen-Shannon distance and agglomerative clustering

Given two distributions P and Q, the JSD is defined as $JSD(P, Q) = \sqrt{\frac{1}{2}[D_{KL}(P||R) + D_{KL}(Q||R)]}$, where $R = (P + Q)/2$ and $D_{KL}$ is the Kullback-Leibler divergence. For a better description of the distances between distributions, an agglomerative hierarchical clustering algorithm was used [66]. Briefly, the clustering method consists in recursively cluster two items at a time. At the beginning, each item defines its own cluster and two most similar items are then clustered. Next, the process is repeated for most similar items or clusters until forming a single cluster. In our case, the JSD was used as a similarity between two languages either from an orthographic or phonetic perspective.

## Communities and $F1$-score

For multiplex networks we adjust the $F1$-score [67] defined in Ref. [68] as follows. Given two collections of communities $\mathcal{S}^{\alpha}$ and $\mathcal{S}^{\beta}$ of layers $\alpha$ and $\beta$, respectively. We define the $F1^*$-score as:

$$F1^*(\mathcal{S}^{\alpha}, \mathcal{S}^{\beta}) = \frac{1}{2}(F1_{\alpha}(\mathcal{S}^{\alpha}, \mathcal{S}^{\beta}) + F1_{\beta}(\mathcal{S}^{\beta}, \mathcal{S}^{\alpha})),$$

where $F1$ represents the average $F1$-score of a reconstructed community with respect to the best match in the opposite layer [68]. It is important to notice that $F1_{\alpha}$ and $F1_{\beta}$ are well defined, since the node sets are the same in both layers ($V^{[\alpha]} = V^{[\beta]}$). The $F1^*$-score between two collections of communities can be interpreted as the degree of similarity between them. For the Louvain method [63] and $F1$-score implementation, we use NetworKit [69].

## Jaccard index

For each node $i$, the local overlap [62] between two layers $\alpha$ and $\beta$ is defined as the total number of nodes such that they are neighbors of node $i$ in both layer $\alpha$ and layer $\beta$:

$$o_i^{[\alpha, \beta]} = \sum_{j=1}^{N} a_{ij}^{[\alpha]} a_{ij}^{[\beta]}.$$

The local overlap can be normalized to have a bounded measure which indicates the similarity

of neighbors of nodes across the layers, obtaining the Jaccard index:

$$J_i^{[\alpha,\beta]} = \frac{o_i^{[\alpha,\beta]}}{\sum\limits_{j=1}^{N}(a_{ij}^{[\alpha]} + a_{ij}^{[\beta]} - a_{ij}^{[\alpha]}a_{ij}^{[\beta]})} \,.$$

The average Jaccard index is constructed by simply considering the total number of nodes in the network.

## Supporting information

**S1 File.**
(PDF)

## Author Contributions

**Conceptualization:** Bibiana Obregón-Quintana, Lev Guzmán-Vargas.

**Data curation:** Pablo Lara-Martínez, C. F. Reyes-Manzano, Irene López-Rodríguez.

**Formal analysis:** Pablo Lara-Martínez, Lev Guzmán-Vargas.

**Investigation:** Bibiana Obregón-Quintana.

**Methodology:** Bibiana Obregón-Quintana, Lev Guzmán-Vargas.

**Project administration:** Lev Guzmán-Vargas.

**Software:** Pablo Lara-Martínez, C. F. Reyes-Manzano, Irene López-Rodríguez.

**Supervision:** Bibiana Obregón-Quintana.

**Validation:** Pablo Lara-Martínez.

**Writing – original draft:** Lev Guzmán-Vargas.

**Writing – review & editing:** Lev Guzmán-Vargas.

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
