## [Decision Letter · Decision Letter 0]

4 Apr 2022

PONE-D-21-38860A multiplex analysis of phonological and orthographicnetworksPLOS ONE

Dear Dr. Guzmán-Vargas,

Thank you for submitting your manuscript to PLOS ONE. After careful consideration, we feel that it has merit but does not fully meet PLOS ONE’s publication criteria as it currently stands. Therefore, we invite you to submit a revised version of the manuscript that addresses the points raised during the review process.

In particular, please consider the suggestions and comments made by the Reviewer #2 as they could strengthen and make more clear your manuscript. 

We look forward to receiving your revised manuscript.

Kind regards,

Irene Sendiña-Nadal

Academic Editor

PLOS ONE

Journal Requirements:

“This work was partially supported by COFAA-IPN, EDI-IPN, and Conacyt-M´exico”

“This work was partially supported by programs EDI and COFAA from Instituto Politécnico Nacional and Consejo Nacional de Ciencia y Tenología, México. No additional external funding was received for this study. The funders had no role in study design, data collection and analysis, decision to publish, or preparation of the manuscript.”

Reviewers' comments:

Reviewer's Responses to Questions

**Comments to the Author**

1. Does the manuscript adhere to the experimental procedures and analyses described in the Registered Report Protocol?

If the manuscript reports any deviations from the planned experimental procedures and analyses, those must be reasonable and adequately justified.

Reviewer #1: Yes

Reviewer #2: Yes

2. If the manuscript reports exploratory analyses or experimental procedures not outlined in the original Registered Report Protocol, are these reasonable, justified and methodologically sound?

A Registered Report may include valid exploratory analyses not previously outlined in the Registered Report Protocol, as long as they are described as such.

Reviewer #1: Yes

Reviewer #2: Yes

3. Are the conclusions supported by the data and do they address the research question presented in the Registered Report Protocol?

The manuscript must describe a technically sound piece of scientific research with data that supports the conclusions. The conclusions must be drawn appropriately based on the research question(s) outlined in the Registered Report Protocol and on the data presented.

Reviewer #1: Yes

Reviewer #2: Yes

4. Have the authors made all data underlying the findings in their manuscript fully available?

Reviewer #1: Yes

Reviewer #2: Yes

5. Is the manuscript presented in an intelligible fashion and written in standard English?

Reviewer #1: Yes

Reviewer #2: Yes

6. Review Comments to the Author

Please use the space provided to explain your answers to the questions above. (Please upload your review as an attachment if it exceeds 20,000 characters)

Reviewer #1: The authors analyze the topological properties of phonological and orthographic networks of 12 languages by adopting a multiplex formalism. They perform a comprehensive analysis of these graphs and uncover interesting patterns of language use.

Strength

* The report is well-prepared, and the analysis is comprehensive.

Weakness

* The figures and captions are out of place. The authors should consider redesigning all the figures to fit within the text, even if it involves splitting the figures into separate subfigures.

I recommend the authors fix the figures (see above) as a minor revision to make this fit for publication.

Reviewer #2: The article entitled “A multiplex analysis of phonological and orthographic networks” explores a multiplex representation of words based on orthographic or phonological similarity to evaluate their structure. The authors perform this study in 12 languages, comparing their network metrics. I do consider this article of great interest as it explores the frontier between written and oral communication by using the novel approach of multiplex networks analysis. Furthermore, this approach may be of great interest in some applications related to Natural Language Processing or Automatic Speech Recognition.

My main criticism of the article is that it seems that a series of very interesting analyzes have been carried out but without a clear objective. For example, in the abstract, it is said: “We report that from the analysis of topological properties of networks, there are different levels of local and global similarity when comparing…” This result somehow would be expected. The interesting question would be univocally framing those results with other linguistic research, phylogenetic approaches, information theory or sociology hypothesis of language. I believe that the article has great potential, but it should make it more straightforward about the hypotheses, objectives, and substantial contributions beyond an exciting series of analyses.

Additionally, I would like to propose the following comments:

Introduction.

Some sentences are vague, e.g. “The complexity of natural language has been studied from different perspectives of scientific research”. This kind of sentence can be more concrete and explicative, including references to previous works.

1. Line 9-13: This should be referenced.

2. Line 13.15: However, it was later fitted to a Weibull distribution.

3. Related to ref 5 and the relationship between ortographic and phonology I would suggest to consider the reference: Torre, I. G., Luque, B., Lacasa, L., Kello, C. T., & Hernández-Fernández, A. (2019). On the physical origin of linguistic laws and lognormality in speech. Royal Society open science, 6(8), 191023; where the authors have addressed this kind of statistical regularities of language.

4. The relationship between orthographic and phonology networks could be of great interest to computer science, where automatic speech recognition systems may fail to transcribe into written text. This somehow could be related to the transfer learning between languages: e.g. a neural network that can recognize a language could work well in another language depending on the similarity of the multiplex network.

5. Limitation of ortographic translation of Damerau-Levenshtein. The orthographic transcription is, in some way, a discrete representation of the spoken word, but the orality is a continuum spectrum. In linguistics, the vowels are represented in the IPA vowel trapezium, some of them physically closer to others. The Damerau-Levenshtein distance does not recover this fact.

Material and methods, and Results

6. Degree distributions are fitted with a Weibull-type function. This hypothesis should be supported by literature and compared to another type of distribution function, e.g. power-law or exponential. 

7. The fitting to a probability distribution function, particularly a power law, has been a hot topic in recent years, so it should be rigorously addressed. The article should include the methodology used, the estimators and the goodness of fit. Some useful references:

• Clauset, A., Shalizi, C. R., & Newman, M. E. (2009). Power-law distributions in empirical data. SIAM review, 51(4), 661-703.

• Navas-Portella, V., González, Á., Serra, I., Vives, E., & Corral, Á. (2019). Universality of power-law exponents by means of maximum-likelihood estimation. Physical Review E, 100(6), 062106.

8. Line 144. The Pearson correlation test hypothesis should be revisited. It usually requires a normal distribution of the values, which seems to be not happening, at least for the degree.

9. Material and methods should be extended to detail the mathematical approaches to fit the distribution and goodness of fit analysis.

Discussion and Conclusions. 

There is no apparent difference between the message written in the discussion and the one written in the conclusion section. I would suggest being more specific or merging both sections.

Reproducibility of the article. 

While the data has been shared, I would recommend to also sharing the scripts used during calculations.

7. PLOS authors have the option to publish the peer review history of their article (what does this mean?). If published, this will include your full peer review and any attached files.

Reviewer #1: No

Reviewer #2: No

---

## [Author Response · Author response to Decision Letter 0]

31 May 2022

Response to Reviewers

Reviewer 1

1. The authors analyze the topological properties of phonological and ortho-

graphic networks of 12 languages by adopting a multiplex formalism. They

perform a comprehensive analysis of these graphs and uncover interesting

patterns of language use.

Strength * The report is well-prepared, and the analysis is comprehensive.

Weakness * The figures and captions are out of place. The authors should

consider redesigning all the figures to fit within the text, even if it involves

splitting the figures into separate subfigures.

I recommend the authors fix the figures (see above) as a minor revision to

make this fit for publication.

Response: Thank for your comments and suggestions. We have

redesigned all the figures as suggested.

Reviewer 2

1. My main criticism of the article is that it seems that a series of very

interesting analyzes have been carried out but without a clear objective.

For example, in the abstract, it is said: “We report that from the analysis

of topological properties of networks, there are different levels of local and

global similarity when comparing. . . ” This result somehow would be ex-

pected. The interesting question would be univocally framing those results

with other linguistic research, phylogenetic approaches, information the-

ory or sociology hypothesis of language. I believe that the article has great

potential, but it should make it more straightforward about the hypotheses,

objectives, and substantial contributions beyond an exciting series of anal-

yses.

Response: We welcome comments on the submitted version.

These observations have been very constructive and are greatly

appreciated. We have revised and added corrections according

to the opinions expressed. Particularly, we have clarified the

1

scope of the study in terms of objectives, hypotheses and con-

tributions.

2. Additionally, I would like to propose the following comments:

Introduction.

Some sentences are vague, e.g. “The complexity of natural language has

been studied from different perspectives of scientific research”. This kind

of sentence can be more concrete and explicative, including references to

previous works.

Response: We have re-written these sentences in a more detailed

way and added citations.

3. Line 9-13: This should be referenced.

Response: We have added the corresponding citations.

4. Line 13.15: However, it was later fitted to a Weibull distribution.

Response: Thank you for the annotation. In previous reports,

the case of phonological networks constructed from similarities

in sounds led to truncated power law type degree distributions

(we have added ref. ([20])). For our part, the similarity is based

entirely on the Damerau-Levenhstein distance, which, although

relatively easier to calculate, tends to identify greater similarity

between words, hence there is a difference between the functions

that best approximate the distributions.

5. Related to ref 5 and the relationship between ortographic and phonology I

would suggest to consider the reference: Torre, I. G., Luque, B., Lacasa,

L., Kello, C. T., Hern ´andez-Fern ´andez, A. (2019). On the physical origin

of linguistic laws and lognormality in speech. Royal Society open science,

6(8), 191023; where the authors have addressed this kind of statistical

regularities of language.

Response: Thank you very much for bringing us closer to the

work of Torre et. al, which is very interesting in the context of

the universality and origin of linguistic laws. We have added the

corresponding citation.

6. The relationship between orthographic and phonology networks could be

of great interest to computer science, where automatic speech recognition

systems may fail to transcribe into written text. This somehow could be

related to the transfer learning between languages: e.g. a neural network

that can recognize a language could work well in another language depend-

ing on the similarity of the multiplex network.

Response: We appreciate this interesting suggestion on the use-

fulness of phonetic-orthographic networks. We have added a few

sentences related to this as motivation in the Introduction (see

lines 50-55)

2

7. Limitation of ortographic translation of Damerau-Levenshtein. The or-

thographic transcription is, in some way, a discrete representation of the

spoken word, but the orality is a continuum spectrum. In linguistics, the

vowels are represented in the IPA vowel trapezium, some of them physi-

cally closer to others. The Damerau-Levenshtein distance does not recover

this fact.

Response: We agree that the similarity based on the Damerau-

Levenhstein distance has limitations that impact especially in

phonetic network, where this affinity is overestimated because

the discretization of sounds leads to loss of structure. We have

added a few sentences pointing out these limitations.

8. Material and methods, and Results

Degree distributions are fitted with a Weibull-type function. This hypoth-

esis should be supported by literature and compared to another type of

distribution function, e.g. power-law or exponential.

Response: We have extended our description and justification

of fitting with a Weibull function (see lines xx-xx)

9. The fitting to a probability distribution function, particularly a power law,

has been a hot topic in recent years, so it should be rigorously addressed.

The article should include the methodology used, the estimators and the

goodness of fit. Some useful references: • Clauset, A., Shalizi, C. R.,

Newman, M. E. (2009). Power-law distributions in empirical data. SIAM

review, 51(4), 661-703. • Navas-Portella, V., Gonz ´alez, ´A., Serra, I.,

Vives, E., Corral, ´A. (2019). Universality of power-law exponents by

means of maximum-likelihood estimation. Physical Review E, 100(6),

062106.

Response: We appreciate the suggestions for a more rigorous

justification of the selection of the best fit to the distributions.

We have added the corresponding citations and included a de-

scription of the tests used.

10. Line 144. The Pearson correlation test hypothesis should be revisited. It

usually requires a normal distribution of the values, which seems to be not

happening, at least for the degree.

Response: Thank you for the observation of the distribution

requirement. We have reviewed the presence of correlations in

terms of Spearman rank correlation coefficient. Unlike the case

of Pearson’s correlation, it does not require the distributions to

be normal. The results are relatively similar.

11. Material and methods should be extended to detail the mathematical ap-

proaches to fit the distribution and goodness of fit analysis.

Response: We have added details of our procedure to fit the dis-

tributions and the goodness of fit analysis (see changes in lines

309-318)

3

12. Discussion and Conclusions.

There is no apparent difference between the message written in the discus-

sion and the one written in the conclusion section. I would suggest being

more specific or merging both sections.

Response: We have reorganized these parts and merged both

Sections

13. Reproducibility of the article.

While the data has been shared, I would recommend to also sharing the

scripts used during calculations.

Response: The scripts of the most representative calculations

are now shared in the FigShare link.

---

## [Decision Letter · Decision Letter 1]

23 Jun 2022

PONE-D-21-38860R1

A multiplex analysis of phonological and orthographicnetworks

PLOS ONE

Dear Dr. Guzmán-Vargas,

Thank you for submitting your manuscript to PLOS ONE. After careful consideration, we feel that it has merit but does not fully meet PLOS ONE’s publication criteria as it currently stands. Therefore, we invite you to submit a revised version of the manuscript that addresses the points raised during the review process.

Although the Authors made a thorough effort to comply with the Reviewer 2's criticisms, there are still some pointed issues which need to be addressed regarding the methodologies and analysis of the empirical data used, in particular the characterisation of the network degree distribution using the least-squares method, which is not recommended at all when the distributions present large fluctuations in their tail. The results of the fittings with most convenient methods to the two already used distributions (or other potential hypothesis) should be provided.   

We look forward to receiving your revised manuscript.

Kind regards,

Irene Sendiña-Nadal

Academic Editor

PLOS ONE

Reviewers' comments:

Reviewer's Responses to Questions

**Comments to the Author**

1. Does the manuscript adhere to the experimental procedures and analyses described in the Registered Report Protocol?

If the manuscript reports any deviations from the planned experimental procedures and analyses, those must be reasonable and adequately justified.

Reviewer #1: Yes

Reviewer #2: Partly

2. If the manuscript reports exploratory analyses or experimental procedures not outlined in the original Registered Report Protocol, are these reasonable, justified and methodologically sound?

A Registered Report may include valid exploratory analyses not previously outlined in the Registered Report Protocol, as long as they are described as such.

Reviewer #1: Yes

Reviewer #2: Yes

3. Are the conclusions supported by the data and do they address the research question presented in the Registered Report Protocol?

The manuscript must describe a technically sound piece of scientific research with data that supports the conclusions. The conclusions must be drawn appropriately based on the research question(s) outlined in the Registered Report Protocol and on the data presented.

Reviewer #1: Yes

Reviewer #2: Partly

4. Have the authors made all data underlying the findings in their manuscript fully available?

Reviewer #1: Yes

Reviewer #2: Yes

5. Is the manuscript presented in an intelligible fashion and written in standard English?

Reviewer #1: Yes

Reviewer #2: Yes

6. Review Comments to the Author

Please use the space provided to explain your answers to the questions above. (Please upload your review as an attachment if it exceeds 20,000 characters)

Reviewer #1: The revised manuscript addresses the concerns raised by R2. However, the issues raised by R1 regarding the misplaced figures still exists. This could be an artifact of TeX-diff, in which case, the authors should ignore the above concerns.

Reviewer #2: First of all, I would like to thank the authors for the time and consideration taken in responding to these suggestions.

Although, in general, the authors have reviewed the comments and worked on the article, I have doubts about one of the points: the adjustments of the network degree distribution. As was stated in the previous version, there has been a lot of research and discussion on this topic during the last decade. Therefore, the fit of a degree distribution by the least-squares method is not recommended at all ((A Clauset · 2007, 2009), Anna Deluca & Álvaro Corral, 2015; among others).

One option is applying the maximum likelihood method to adjust the parameters and detect the lower cut-off points, but other statistical methods are available.

Furthermore, different candidate distributions should be considered when the slopes are not very steep (as seems to be the case), typically Weibull, lognormal, power law, gamma or exponential. MLE numerical results should be provided with the values of loglikelihood, BIC or AIC. It should be considered that some of these distributions have a different number of variables. The best candidate distribution should be selected with loglikelihood, AIC or BIC. Finally, the goodness of fit should also be provided. The results of those fitting should be added to the main document or supplementary material, and ideally, the programming scripts should also be provided.

Chattopadhyay, S., Murthy, C. A., & Pal, S. K. (2014). Fitting truncated geometric distributions in large-scale real-world networks. Theoretical Computer Science, 551, 22-38.

Clauset, A., Shalizi, C. R., & Newman, M. E. (2009). Power-law distributions in empirical data. SIAM Review, 51(4), 661-703.

7. PLOS authors have the option to publish the peer review history of their article (what does this mean?). If published, this will include your full peer review and any attached files.

Reviewer #1: No

Reviewer #2: No

---

## [Author Response · Author response to Decision Letter 1]

12 Aug 2022

Response to Reviewers

Reviewer 1

1. The revised manuscript addresses the concerns raised by R2. However,

the issues raised by R1 regarding the misplaced figures still exists. This

could be an artifact of TeX-diff, in which case, the authors should ignore

the above concerns.

Response: We have corrected the Figures.

Reviewer 2

1. Although, in general, the authors have reviewed the comments and worked

on the article, I have doubts about one of the points: the adjustments of

the network degree distribution. As was stated in the previous version,

there has been a lot of research and discussion on this topic during the

last decade. Therefore, the fit of a degree distribution by the least-squares

method is not recommended at all (A Clauset · 2007, 2009, Anna Deluca

 ´Alvaro Corral, 2015; among others). One option is applying the maximum

likelihood method to adjust the parameters and detect the lower cut-off

points, but other statistical methods are available. Furthermore, different

candidate distributions should be considered when the slopes are not very

steep (as seems to be the case), typically Weibull, lognormal, power law,

gamma or exponential. MLE numerical results should be provided with the

values of loglikelihood, BIC or AIC. It should be considered that some of

these distributions have a different number of variables. The best candidate

distribution should be selected with loglikelihood, AIC or BIC. Finally, the

goodness of fit should also be provided. The results of those fitting should

be added to the main document or supplementary material, and ideally,

the programming scripts should also be provided.

Response: We appreciate the suggestion to explore in more de-

tail the problem of fitting empirical degree distributions. We

have adopted this strategy and several functions have been con-

sidered. Our approach has been based, as suggested, on the

AIC and BIC indices, to identify the best fit. In addition, we

have used the reason methodology proposed by A. Clauset et

al. to confirm that the candidate distribution is better com-

pared to the other distributions (see lines 134-157 of the main

article and the Supplementary Material in FigShare ( https:

//doi.org/10.6084/m9.figshare.14668593 ). Additional details,

such as the results of the fittings, test values and programming

scripts performed are also shown in the material available on-line.

---

## [Decision Letter · Decision Letter 2]

1 Sep 2022

A multiplex analysis of phonological and orthographicnetworks

PONE-D-21-38860R2

Dear Dr. Guzmán-Vargas,

We’re pleased to inform you that your manuscript has been judged scientifically suitable for publication and will be formally accepted for publication once it meets all outstanding technical requirements.

Kind regards,

Irene Sendiña-Nadal

Academic Editor

PLOS ONE

Additional Editor Comments (optional):

Reviewers' comments:

Reviewer's Responses to Questions

**Comments to the Author**

1. Does the manuscript adhere to the experimental procedures and analyses described in the Registered Report Protocol?

If the manuscript reports any deviations from the planned experimental procedures and analyses, those must be reasonable and adequately justified.

Reviewer #2: Yes

2. If the manuscript reports exploratory analyses or experimental procedures not outlined in the original Registered Report Protocol, are these reasonable, justified and methodologically sound?

A Registered Report may include valid exploratory analyses not previously outlined in the Registered Report Protocol, as long as they are described as such.

Reviewer #2: Yes

3. Are the conclusions supported by the data and do they address the research question presented in the Registered Report Protocol?

The manuscript must describe a technically sound piece of scientific research with data that supports the conclusions. The conclusions must be drawn appropriately based on the research question(s) outlined in the Registered Report Protocol and on the data presented.

Reviewer #2: Yes

4. Have the authors made all data underlying the findings in their manuscript fully available?

Reviewer #2: Yes

5. Is the manuscript presented in an intelligible fashion and written in standard English?

Reviewer #2: Yes

6. Review Comments to the Author

Please use the space provided to explain your answers to the questions above. (Please upload your review as an attachment if it exceeds 20,000 characters)

Reviewer #2: The authors have addressed all recommendations and included all requested information about the statistical analysis and fittings. Besides, they have included additional details in Supplementary Information. Finally, I just want to congratulate the authors on this interesting scientific paper and acknowledge their efforts.

7. PLOS authors have the option to publish the peer review history of their article (what does this mean?). If published, this will include your full peer review and any attached files.

Reviewer #2: No

---

## [Editor Report · Acceptance letter]

6 Sep 2022

PONE-D-21-38860R2 

A multiplex analysis of phonological and orthographic networks 

Dear Dr. Guzmán-Vargas:

I'm pleased to inform you that your manuscript has been deemed suitable for publication in PLOS ONE. Congratulations! Your manuscript is now with our production department. 

Kind regards, 

on behalf of

Dr. Irene Sendiña-Nadal 

Academic Editor

PLOS ONE